# Simulations of Temperature-Dependent Magnetization in $Fe_xGd_{100-x}$ ($20 \leq x \leq 80$) Alloys

**Oleksandr Pastukh** *[ID], **Dominika Kuźma** *[ID] **and Svitlana Pastukh** [ID]

Institute of Nuclear Physics Polish Academy of Sciences, PL-31342 Kraków, Poland
* Correspondence: oleksandr.pastukh@ifj.edu.pl (O.P.); dominika.kuzma@ifj.edu.pl (D.K.)

**Abstract:** Theoretical calculations of the temperature-dependent magnetization in FeGd alloys were done with the use of Heisenberg-type atomistic spin Hamiltonian and Monte Carlo algorithms. The random allocation of atoms in the desired crystal structure was used for simulations of magnetically amorphous alloys. Performed calculations for the two different crystal structures have shown an important role of coordination number on the observed critical temperature and compensation point. Moreover, the value of the exchange interaction between Fe and Gd sublattices plays a key role in the simulations—an increase in the Fe–Gd exchange constant provides an increase in critical temperature for each concentration of elements, which explains the higher temperature stabilization of Gd moments. It was shown that obtained temperature-dependent magnetization behavior is consistent with experimental observations, which confirms the applicability of the atomic model used to study FeGd or other magnetic alloy structures.

**Keywords:** magnetic alloy; Monte Carlo simulations; magnetization compensation

## 1. Introduction

The investigation of magnetic properties of amorphous and intermetallic alloys consisting of transition metal (TM) and rare-earth (RE) elements brings attention from the first studies at the beginning of the 1960s [1–4]. TM and RE alloys have magnetic moments originating from electrons on the *d* and *f* orbitals, respectively, with strong antiferromagnetic exchange coupling for certain combinations of such elements [1,2,5–8]. Such materials can form complex magnetic structures with a high variation of magnetization, depending on their composition, and have great application potential in data storage, spintronics and optical technologies [9]. Owing to the possibility of high-speed manipulation of the magnetization, such compounds are ideal candidates for magneto-optic recording applications [10,11]. A large magnetocaloric effect in TM-RE materials can be considered for their application in magnetic refrigeration [12,13]. Many different TM-RE alloys are also commonly used as spin-valve [14,15] or spin-orbit torque devices [16].

Among different rare earth elements, gadolinium reveals a ferromagnetic nature at room temperature and is therefore commonly used for the fabrication of alloys with transition metal elements. The FeGd alloys are of special interest in recent experimental and theoretical investigations. The substantial difference in the ordering temperature of Fe and Gd ($T_c$ = 1043 K and $T_c$ = 293 K, respectively), and atomic spin moments values ($\mu_s$ = 2.22 $\mu_B$ and $\mu_s$ = 7.63 $\mu_B$, respectively), together with strong antiferromagnetic exchange interactions, bring unique magnetic characteristics to the alloys of such structures. The magnetization behavior that depends on the composition and morphology of FeGd alloys was investigated previously [10,17–25]. The study of ultrafast magnetization dynamics in FeGd amorphous alloys shows the possibility of manipulating the magnetic order of the two sublattices on the timescale of the exchange interaction, which is of great importance in magnetic recording and information processing [23,24,26]. FeGd alloys also have useful technological applications. Its layers are taken advantage of as magneto-optical memory

media [27] or giant magnetoresistance devices [28]. As an example, a magneto-optical spatial light modulator for 3D holography applications was created on this basis [29]. Therefore, the theoretical study of FeGd alloys could reveal new and interesting features in magnetic behavior, which are suitable for technological applications.

It is known from the literature that amorphous TM-RE alloys are synthesized by various techniques, the main one being based on the rapid quenching of the constituents from the liquid phase. In this case, the atoms become immobilized in the disordered configuration [30]. In FeGd alloys, the large difference in the atomic diameters of Fe and Gd ions provides significant lattice distortion. Random electrostatic fields generated by such disordered structures result in changes in the rare-earth moments' magnitude and direction, which is reflected in the observed magnetic properties. However, to take into account the real size of atoms and ostensible distorted structure, ab initio methods should be applied, which are, in most cases, complicated and demand large computational effort for large systems. On the other hand, the use of random atom allocation in atomistic model simulations can provide a good approximation of amorphous alloy structure and create the possibility to study magnetic properties in a simple way.

In the current study, the theoretical simulation of temperature-dependent magnetization was done for $Fe_xGd_{100-x}$ magnetically amorphous alloys with a broad range of elements composition change ($20 \leq x \leq 80$) within the atomistic spin model included in the Vampire software (York, England) [31]. The applied model allows for the possibility to simulate an alloy structure in which atoms are randomly allocated within the chosen crystal structure and approximate the effect of real amorphous materials. Within the simulations, two different crystal structures were considered, yielding different allocations of TM and RE atoms. The different values of exchange between the iron and gadolinium sublattices were also considered, and the change in critical temperature for different concentrations of TM/RE elements was discussed and compared with the experimental data.

## 2. Materials and Methods

It is important to highlight at the beginning that system generation in Vampire allows for the simulation of alloy material by the random allocation of atoms of different elements in the chosen crystal structure. This allows for the possibility to approximate the effects of real amorphous alloy materials. Since the crystal structure is used as a basis for random alloy formation, a different coordination number in certain structures may influence the simulated magnetic behavior. Therefore, we performed a calculation using two different structures, bcc and hcp, as the counterparts of Fe and Gd bulk materials, respectively. The generation of the alloy system was done by the following procedure. The $n \times n \times n$ cells of the defined crystal lattice (bcc and hcp) were generated with applied periodic boundary conditions in x, y and z dimensions to simulate the effect of bulk structure. Then, the formed crystal was populated with a random distribution of Fe and Gd atoms in the defined concentrations (according to the desired composition of $Fe_xGd_{100-x}$ alloy). In such a way, the random allocation of TM and RE ions in the crystal was used to approximate the effects of an amorphous TM-RE alloy. The simulated structures used in the study are shown in Figure 1 for three exemplary alloy compositions. Let us note that such a theoretical model is confirmed to be successfully used for simulations of simple magnetically amorphous alloys and calculations of their magnetic properties, which are comparable with the experimental data [21].

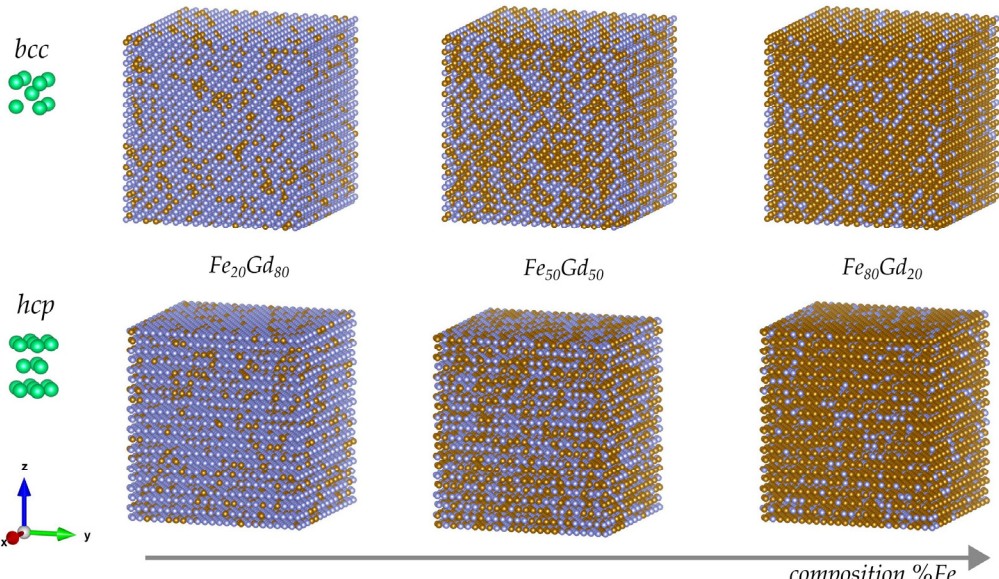

**Figure 1.** Graphical visualization of the simulated structures of FeGd alloys with different compositions and crystal structures used as the basis of random alloy generation (three exemplary concentrations of Fe atoms in alloys are shown). Blue balls correspond to gadolinium used atoms and orange to iron atoms, whereas the green balls shown are the crystal structures.

It is important to note that due to factors such as the coordination number and variations in the relative concentrations of the various materials throughout the alloy sample, the choice of crystal structure is crucial in determining the effective exchange between the Fe and Gd ions. Furthermore, the magnetic characteristics of amorphous alloys are primarily influenced by the strength of exchange interactions between the nearest neighbors [32]. Therefore, in the current study, we investigate the magnetization dependence on the temperature in the $Fe_xGd_{100-x}$ alloy, taking into account the change of the following parameters: a broad range of change of composition of Fe and Gd ions in the alloy (from x = 20 to x = 80), two different types of structures used in the material formation (bcc and hcp) and different values of the exchange interaction energy between atoms. The theoretical study is based on the determination of the critical temperature with the change of the above-mentioned parameters in FeGd alloys.

For the simulation of the *M(T)* behavior, the Heisenberg-type spin Hamiltonian and Monte Carlo (MC) algorithm were applied. The MC algorithm used is based on the so-called trial moves—change of the initial spin direction of random spin to the new trial position and then the calculation of transition probability of its acceptance, based on the energy difference between the old and new position. The process is then repeated until N trial moves—where N is the total number of spins in the system—have been made. Each set of N trial moves represents a single MC step in calculations. The Hinzke–Nowak method [33], which uses a mixture of different trial moves (spin flip, Gaussian and random), is applied in the software [31].

For the simulation of temperature-dependent magnetization, the 20,000 equilibration steps followed by 20,000 averaging steps were applied for reaching the equilibration and the system was then heated in steps of 5.0 K. The complete *M(T)* curve was simulated in the temperature range from 0 K to 1700 K. The Hamiltonian algorithm of the system has the following form:

$$\mathcal{H} = \mathcal{H}_{exc}^{Fe-Fe} + \mathcal{H}_{exc}^{Gd-Gd} + \mathcal{H}_{exc}^{Fe-Gd}, \tag{1}$$

$$\mathcal{H}_{exc} = -\sum_{i \neq j} J_{ij} S_i S_j. \tag{2}$$

The first and the second term in Equation (1) take into account the exchange interaction between iron and gadolinium atoms, respectively, whereas the third term refers to Fe–

Gd exchange interactions. In Equation (2), $J_{ij}$ is the exchange constant describing an interaction between atomic sites $i$ and $j$, and $S_i$ and $S_j$ are unit vectors, denoting the local spin moment directions of sites $i$ and $j$. It is important to note that for the case of Fe–Fe and Gd–Gd interactions, the exchange coupling constant is positive, as it is for the ferromagnetic materials. The $J_{Fe-Fe}$ and $J_{Gd-Gd}$ values were taken as for the corresponding bulk materials, $7.050 \times 10^{-21}$ J/link and $1.280 \times 10^{-21}$ J/link, respectively. Considering the antiferromagnetic interactions between Fe and Gd ions, the corresponding exchange coupling constant was taken as negative. Since the exact value of the $J_{Fe-Gd}$ exchange constantly differs in the reported literature data for similar investigations, we take into account two different values, namely $J_{Fe-Gd} = -1.380 \times 10^{-21}$ J/link, as it was used in [24,34] and the two times larger value $J_{Fe-Gd} = -2.760 \times 10^{-21}$ J/link, as it was in [20]. Let us note that in the performed calculations, the anisotropy energy was also included (the uniaxial for the case of hcp structure and cubic for the case of bcc structure); however, its impact on the total magnetic energy is less significant. The magnetic moments of materials were taken as follows: $\mu_s^{Fe} = 2.22 \ \mu_B$ and $\mu_s^{Gd} = 7.63 \ \mu_B$.

### 3. Results and Discussion

Before the investigation of the temperature-dependent magnetization of the alloys, it made sense to test the applied atomistic model and the magnetic parameters used for bulk iron and gadolinium structures in order to obtain the correct values of corresponding Curie temperature. For that purpose, the corresponding structures (Fe and Gd) were generated with the above magnetic parameters and structure, taken from the literature data [35,36]: bcc crystal structure with a unit cell size of a = 2.866 Å for the iron and hcp structure with a = 3.636 Å and c = 5.783 Å for gadolinium.

The calculated temperature-dependent normalized magnetization (*M/Ms*, hereafter called *M(T)*) curves, are shown in Figure 2. In order to determine the value of the critical temperature, different approximations can be used, such as Curie–Bloch or Kuz'min equations [37]. These relations allow for a good estimation of the Curie temperature for the elemental ferromagnets; however, very often, the magnetic transitions for the material can be smeared and the determination of $T_c$ loses its criticality. Therefore, we used the maximum of the mean susceptibility curve as the method for critical temperature determination (Figure 2). Obtained values $T_c^{Fe} = 1045(5)$ K and $T_c^{Gd} = 295(5)$ K for the investigated materials are in good agreement with the experimental data for bulk iron and gadolinium ferromagnets. In such a way, the model used describes the *M(T)* behavior of bulk Fe and Gd structures well and can be applied for further analysis.

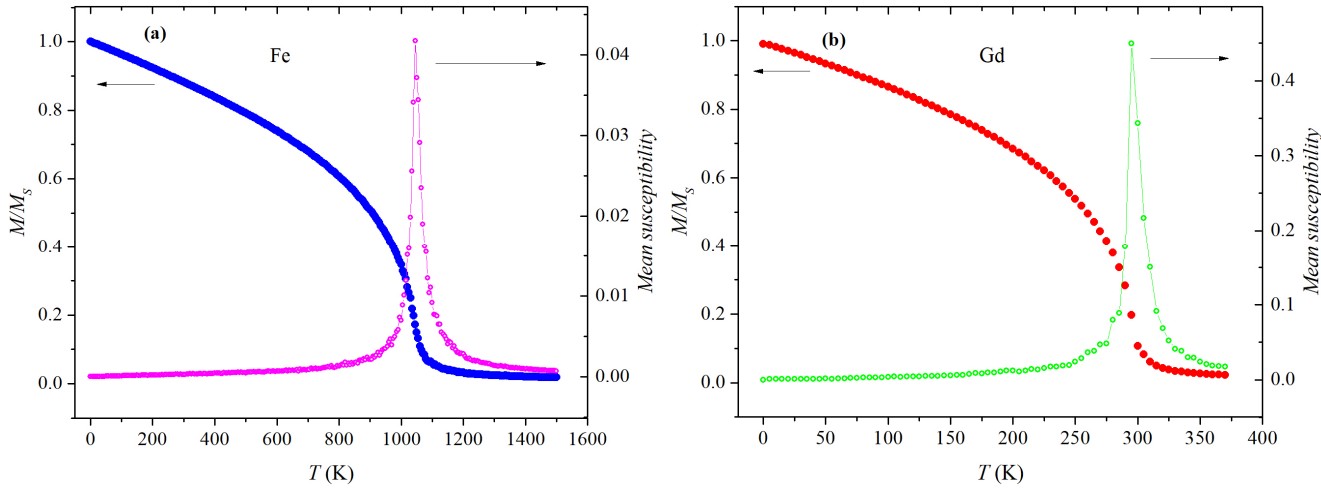

**Figure 2.** Simulations of the dependence of normalized magnetization and mean susceptibility on the temperature for bulk bcc-iron (**a**) and hcp-gadolinium (**b**) materials.

In the next step, the calculations of temperature-dependent magnetization were performed for the $Fe_xGd_{100-x}$ alloys ($20 \leq x \leq 80$) with the exchange interaction $J_{Fe-Gd} = -1.380 \times 10^{-21}$ J/link. The corresponding curves are shown in Figure 3a for the case of the bcc system and in Figure 3b for the hcp system. As can be seen, the change in the composition of alloy strongly influences the magnetic behavior and observed critical temperature. In both cases for the lowest x value, when the concentration of Gd atoms dominates, a sharp decrease of magnetization with temperature is observed and the critical temperature is in the range of $T_c$ for bulk gadolinium. With the increase of Fe concentration (or decrease of Gd concentration) in the alloys, the *M(T)* curves become more smooth and lose their criticality near the $T_c$. The observed ordering temperature rises (see insets of Figure 3a,b), indicating that the major interactions involving Fe moments are of type Fe–Fe or Fe–Gd. Such behavior is observed for the Fe content up to x = 50.

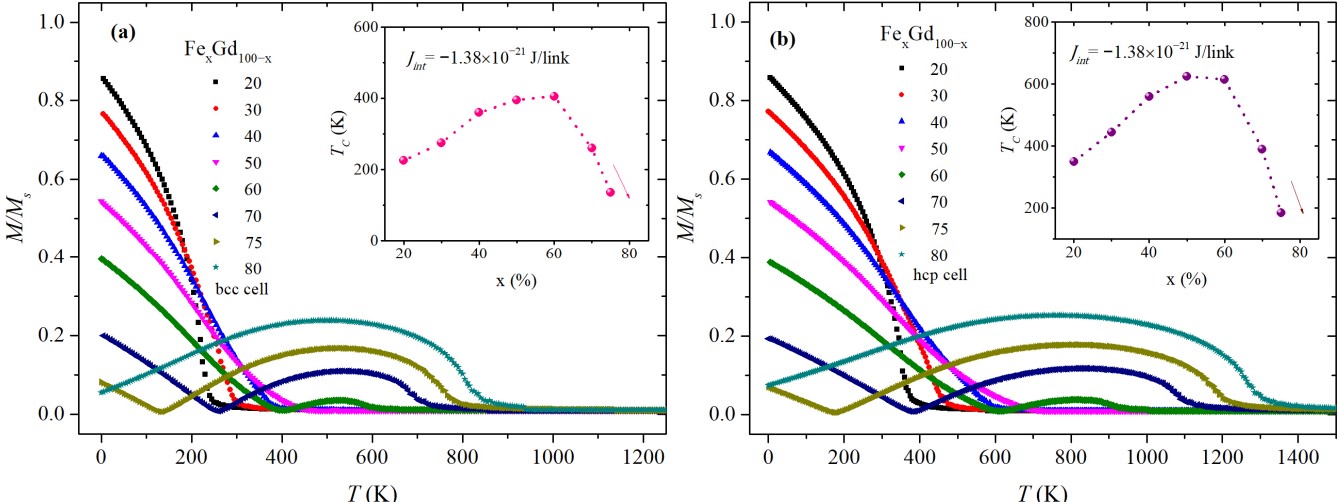

**Figure 3.** Simulations of temperature-dependent normalized magnetization of the $Fe_xGd_{100-x}$ alloys with antiferromagnetic exchange interactions for the bcc (**a**) and hcp (**b**) structures for the value of exchange interaction $J_{Fe-Gd} = -1.380 \times 10^{-21}$ J/link. The insets of the figures show the concentration dependence of the critical temperature, determined from the corresponding *M(T)* curves.

The completely different magnetic behaviors, however, may be seen in the further increase of iron atoms content in the alloys. For the $60 \leq x \leq 75$, the corresponding curves decrease rapidly as the temperature is raised and vanishes at a so-called compensation temperature. With a further increase in temperature, the magnetization reappears with the broad peak and finally decreases to zero at the Curie temperature. The appearance of the compensation point is well-known in transition metal/rare earth alloys [10,11] and is explained by the existence of two sublattices in the structure with different magnetization changes over the temperature [38]. The important role here is played by the Fe–Gd interactions since the magnetization of Gd and Fe sublattices are antiparallel coupled. At zero temperature, the magnetization of the Gd sublattice is higher than that of the Fe atoms; however, it decreases rapidly with the increase of temperature and reaches the iron sublattice magnetization at a certain temperature point. When both magnetizations become equal, they cancel each other out and the compensation is observed with nearly zero net magnetization. For the further increase of temperature, iron magnetization dominates and the increase of magnetization with temperature is observed.

As can be seen from Figure 3, the compensation point shifts toward the lower temperatures with the increase of Fe concentration. This is because the ferromagnetic exchange between iron atoms becomes more dominant than the antiferromagnetic Fe–Gd interactions. The compensation temperature moves close to 0 K for $Fe_{80}Gd_{20}$ and is not observed for the higher x values. The range of Fe:Gd concentrations for which the compensation effect

is observed ($60 \leq x \leq 80$) is in good agreement with the experimental observations [10], which confirms the correctness of the applied theoretical model.

Another observation can be done by comparing *M(T)* curves for the case of bcc and hcp structures, which were used as a basis for the random FeGd alloy simulations. The shape of the curves is similar for the corresponding concentrations of elements, and the concentration dependence of the critical temperature (see insets of Figure 3) has a similar trend. Although the *M(T)* curves are qualitatively similar, the values of corresponding critical temperatures differ and are generally higher for the hcp structure of random alloys. This can be explained by the different coordination number in both structure types (8 for bcc and 12 for hcp), since the number of atoms bonded to the central atom in the alloy determines the effective exchange between the two materials. Furthermore, both types of structures have different spatial variations of the Gd and Fe ions concentrations. These two factors may influence the ferromagnetic Fe–Fe and Gd–Gd interactions, as well as antiferromagnetic Fe–Gd interactions and play a key role in the obtained value of critical temperature.

Since the value of observed critical temperature in the magnetic materials is defined mainly by the strength of the inter-atomic exchange interactions, in the next step, we also performed temperature-dependent simulations for the different value of $J_{Fe-Gd}$. Figure 4 shows the temperature dependence of normalized magnetization with the change of the FeGd alloy composition for bcc and hcp structures, calculated with the exchange interaction value $J_{Fe-Gd} = -2.760 \times 10^{-21}$ J/link. As can be seen, the behavior of *M(H)* curves for different concentrations of elements in the alloy is qualitatively similar to the results obtained for lower Fe–Gd exchange energy. The lower critical temperature was observed for the small Fe content because of dominant gadolinium interactions. With the increase of Fe atoms concentration, the curves become more smooth and the compensation temperature effect appears at the $60 \leq x \leq 80$ in $Fe_xGd_{100-x}$ alloys. The main difference concerns the values of critical temperature, which increase for each concentration of elements in the alloys with the increase of the exchange coupling constant between TM and RE elements (see insets on Figure 4). In this case, because the exchange energy between iron and gadolinium atoms ($J_{Fe-Gd}$) are about two times larger than between gadolinium atoms ($J_{Gd-Gd}$), the stabilization of the Gd magnetic moment appears at a higher temperature. As a result, the critical temperature (including the stabilization temperature) moves to higher values.

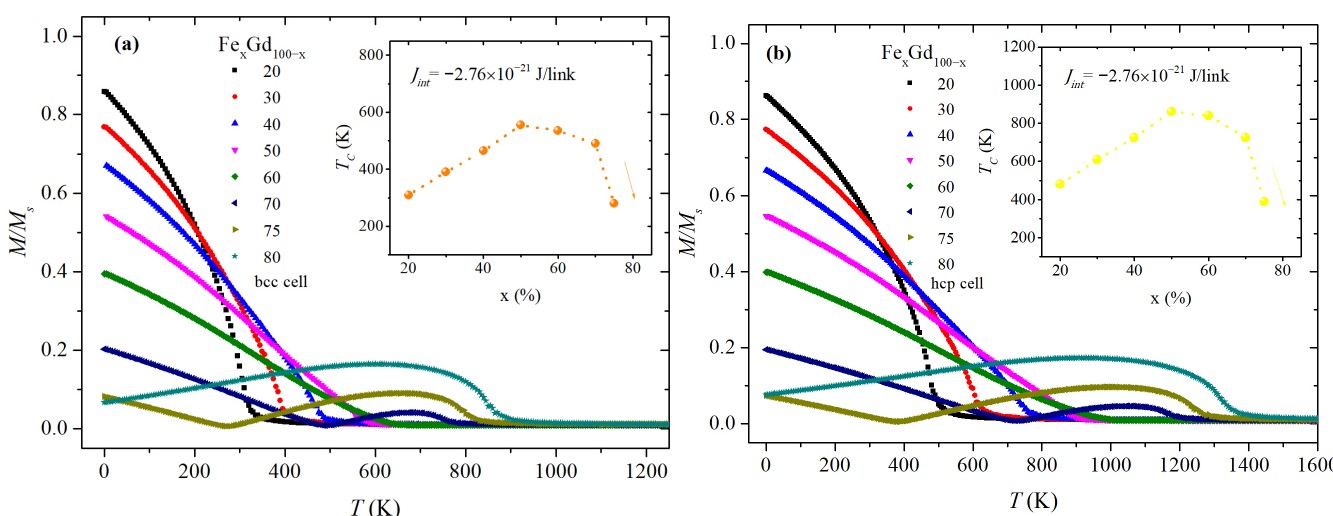

**Figure 4.** Simulations of temperature-dependent normalized magnetization of the $Fe_xGd_{100-x}$ alloys with antiferromagnetic exchange interactions for the bcc (**a**) and hcp (**b**) structures for the value of exchange interaction $J_{Fe-Gd} = -2.760 \times 10^{-21}$ J/link. The insets of the figures show the concentration dependence of the critical temperature, determined from the corresponding *M(T)* curves.

In addition to the performed calculations of *M(T)* dependence, the visualization of spin configuration changes with temperature increases and can be done for the investigated alloys. With the use of the POVRay [39] visualization tool, the magnetization direction of each individual atom in a structure can be simulated. In the absence of an external field, alloys possess only spontaneous magnetization; therefore, the spin projection evolution with the temperature was shown for one exemplary structure (Figure 5). As we can see, the visible antiferromagnetic configuration of Fe and Gd spins, relative to each other, can be seen for the initial state (see Figure 5a). As the temperature rises, thermal energy begins to compete with exchange energy, causing more spins to become disoriented until the temperature reaches the paramagnetic state, where all order is totally lost (see Figure 5b).

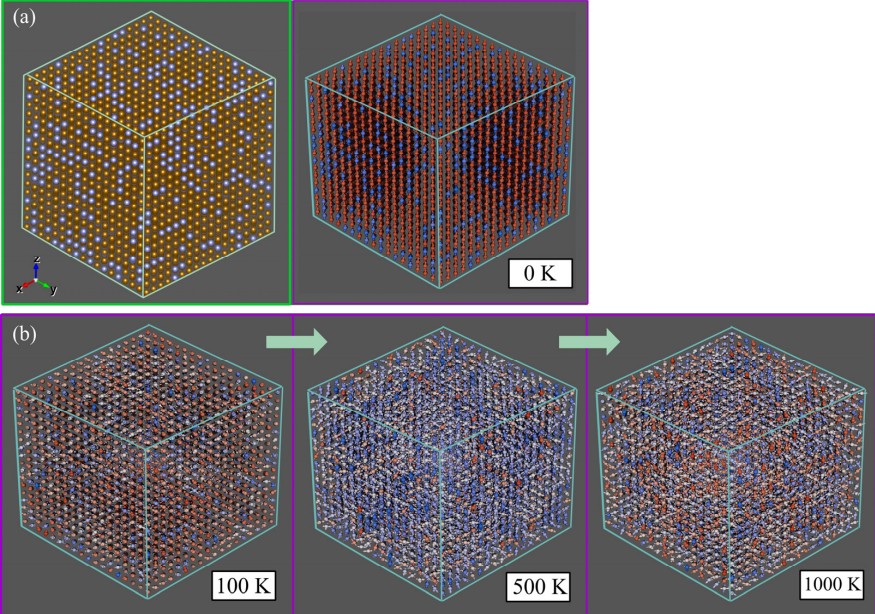

**Figure 5.** The representation of spin configurations of simulated $Fe_{75}Gd_{25}$ (bcc) alloy ($J_{Fe\text{-}Gd}$ = $-1.380 \times 10^{-21}$ J/link) at the initial state (0 K) with antiferromagnetic Fe/Gd order (**a**) and with the temperature increase (**b**). The color changes with the spin projection. The pure red and pure blue colors represent directions opposed to each other. The leftmost image in (**a**) shows the visualization of the simulated structure (color scheme the same as in Figure 1).

As we have shown, the temperature-dependent magnetization in $Fe_xGd_{100-x}$ alloys has a similar trend of change with the concentration of the elements ($20 \leq x \leq 80$) for different magnetic and structural parameters; however, observed critical temperature values may slightly change. The obtained theoretical data can be then compared with the experimentally performed measurements of analogous structures. One of the earliest magnetic investigations of amorphous FeGd alloys prepared by the melt spinning technique was done by Buschow et al. [18]. The author studied the magnetization behavior and determined the critical temperatures for the alloys with low Fe concentration ($30 \leq x \leq 50$). Buschow et al. reported the magnetic ordering temperature for $Fe_{32}Gd_{68}$ to be near room temperature (RT) for $Fe_{40}Gd_{60}$—above RT and for $Fe_{50}Gd_{50}$—above 400 K. These values are in good agreement with the performed simulations for the alloys with corresponding compositions for the bcc structure with exchange energy value $J_{Fe-Gd}$ = $-1.380 \times 10^{-21}$ J/link (theoretically obtained values are $T_c(Fe_{30}Gd_{70})$ = 275(5) K, $T_c(Fe_{40}Gd_{60})$ = 360(5) K and $T_c(Fe_{50}Gd_{50})$ = 395(5) K). The wide range of elements concentrations in amorphous FeGd alloys, prepared with the melt quenching method, was studied by Yano et al. [40]. The comparison between the corresponding calculated and experimental data is shown in Figure 6b and Table 1. The reported data show a monotonous increase of critical temperature with iron concentration up to about 60% and then a slight decrease of $T_c$ is observed in Fe rich

region. The similar trend of $T_c$ changes with temperature is also observed in the performed calculations; however, the results are quantitatively different for some calculated samples. On the other side, there is a good agreement between the experimental data for the amorphous structure with the data obtained for the simulated FeGd alloy with bcc structure and $J_{Fe-Gd} = -2.760 \times 10^{-21}$ J/link in the region of iron concentration between 30 and 60%. The corresponding values are highlighted in Table 1. The little divergence between theoretical and experimental data for such a sample starts at high iron concentrations (x > 60%); however, both dependencies have the same trend. On the other side, the critical temperature data in this Fe:Gd ratio region seems to be in better agreement with the results for the crystalline FeGd alloy, as reported by Vickery et al. [3] (see the rightmost column in Table 1). Here we can suppose that a dominant amount of Fe atoms in the modeled sample can behave similarly to the crystalline FeGd structure since the amorphous alloys are simulated in the software used by randomly allocating atoms in the crystal structure.

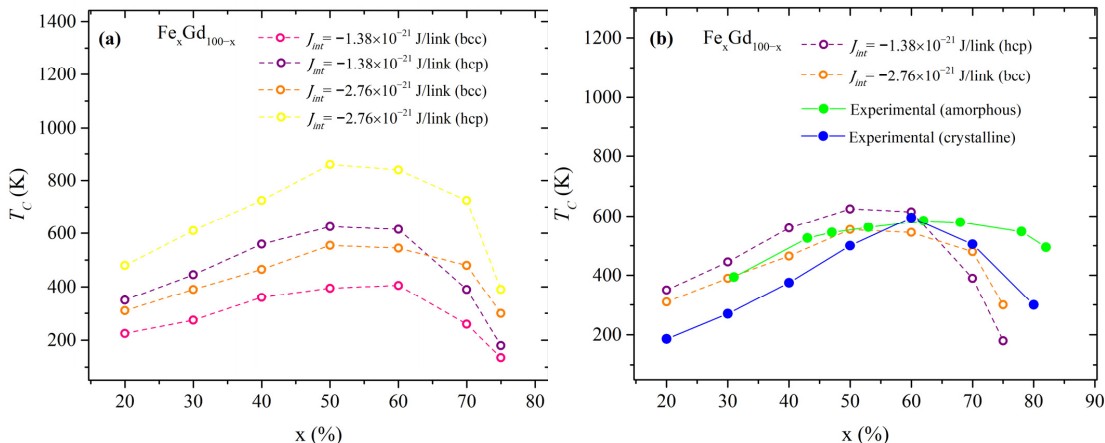

**Figure 6.** Concentration dependence of the critical temperature of the $Fe_xGd_{100-x}$ alloys with bcc and hcp structure for different values of the Fe–Gd exchange energy (**a**) and comparison of theoretical data (open circles) with experimental data (full circles) from Ref. [40] (amorphous structure) and Ref. [3] (crystal structure) (**b**).

**Table 1.** The values of critical temperature obtained from calculations for different $Fe_xGd_{100-x}$ alloy compositions and magnetic parameters. The following notation was used: $J_1 = J_{Fe-Gd} = -1.380 \times 10^{-21}$ J/link and $J_2 = J_{Fe-Gd} = -2.760 \times 10^{-21}$ J/link. The experimental data were extrapolated for the corresponding concentrations from the concentration–temperature relationship presented in Ref. [40] for the amorphous structure and Ref. [3] for the crystal structure.

| | $T_c$ (K) | | | | | |
| :---: | :---: | :---: | :---: | :---: | :---: | :---: |
| $x$ (%) | bcc; $J_1$ | hcp; $J_1$ | bcc; $J_2$ | hcp; $J_2$ | Exp. Amorphous | Exp. Crystal |
| 20 | 225(5) | 350(5) | 310(5) | 480(5) | — | ~186 |
| 30 | 275(5) | 445(5) | 390(5) | 610(5) | ~382 | ~270 |
| 40 | 360(5) | 560(5) | 465(5) | 725(5) | ~495 | ~375 |
| 50 | 395(5) | 625(5) | 555(5) | 860(5) | ~553 | ~500 |
| 60 | 405(5) | 615(5) | 545(5) | 840(5) | ~575 | ~596 |
| 70 | 260(5) | 390(5) | 480(5) | 725(5) | ~570 | ~505 |
| 75 | 135(5) | 180(5) | 305(5) | 390(5) | ~557 | ~392 |

## 4. Conclusions

To summarize, the theoretical model used can be successfully applied for the simulation and calculation of temperature-dependent magnetization of TM-RE alloys. The obtained data of critical temperature seems to be comparable with the experimental measurements, taking into account the different compositions of elements in the amorphous alloys. As it was shown, the change in the concentration of elements plays an important role in observed magnetization behavior and provides the appearance of the compensation temperature for the alloys with iron content of $60 \leq x \leq 75$, which is consistent with the experimental observations. Considering the two different crystal structures used in random alloy formation, we have demonstrated the important role of the coordination number, which may modify the spatial variations of the Gd and Fe ions concentrations and, as a result, the effective exchange between two sublattices. In addition, it was shown that the critical temperature value obtained from the calculations increases for each concentration of elements in the alloys with the increase of the exchange coupling constant between TM and RE elements, which is explained by the higher temperature stabilization of Gd magnetic moments. Performed calculation can also be considered as an alternative way for the theoretical study of alloy structures in contrast to the simulations based on the first-principle methods. Note that the obtained results can be of interest for further theoretical analysis of transition metal/rare earth alloy structures and reveal new specific features in magnetic behaviors that are important for practical applications.

**Author Contributions:** Conceptualization, O.P., D.K. and S.P.; methodology, O.P., D.K. and S.P.; software, O.P.; visualization, O.P.; investigation, O.P., D.K. and S.P.; data curation, O.P.; writing—original draft preparation, O.P., D.K. and S.P.; writing—review and editing, O.P., D.K. and S.P.; supervision, O.P.; funding acquisition, D.K. All authors have read and agreed to the published version of the manuscript.

**Funding:** The numerical calculations were performed at Poznan Supercomputing and Networking Center (Grant No. 424).

**Conflicts of Interest:** The authors declare no conflict of interest.

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
