# Peer review of "Simulations of Temperature-Dependent Magnetization in FexGd100−x (20 ≤ x ≤ 80) Alloys"

_2673-8724, doi:10.3390/magnetism3010004_

Round 1

Reviewer 1 Report

This work looks good. The  temperature dependent magnetic property is very important, but it is difficult for its simulation based on ab initio calculations. The present work is a available approach. The exchange interaction parameters are key role in total simulations. Serval comments:

1. how many size in the simulations.

2. The large atomic radius difference exists between Fe and Gd, which could results in large lattice distortion. Authors should give the corresponding discussion. 

Reviewer 2 Report

In this work, the theoretical calculations of the temperature-dependent magnetization in FeGd alloys were done with the use of Heisenberg-type atomistic spin Hamiltonian and Monte Carlo algorithm. The work reported in the manuscript is systematic and detailed. I suppose the paper will give us some valuable recommendations. To the reviewer's opinion, this paper can be considered in this journal. But before that, here are some details and problems that need to be enriched.

First, in this manuscript, the random allocation of atoms in the desired crystal structure was used for simulations of magnetically amorphous alloys. My question is how to verify the authenticity of random allocation. Does it exist in the experiment?  

Moreover, authors claim that the obtained critical temperature and compensation point dependence on the concentration of the elements in the alloy was shown to be in good agreement with experimental observations. As a general reader, I cannot find the same results easily between this work and the references. Maybe more detailed comparisons and information should be provided.

Reviewer 3 Report

This manuscript has studied the temperature dependent magnetic properties of amorphous Fe_xGd_{100-x} on various compositions with atomistic simulations. Key magnetic parameters such as the Curie temperature and the compensation point are computed for two different crystal structures and different anti-ferromagnetic exchange couplings between Gd and Fe atoms. While the study is based on well established methods and the results were benchmarked with past experimental work, my main concern is the lack of key messages or insights from this study, at least not being emphasized or organized well in the manuscript. Here are a few points for the authors to consider:

  1. This comment is about the abstract and the conclusion part. It is not clear what the readers can learn from this work based on the abstract and the conclusion. The abstract only states what study has been done and the results are agreeing with experiments (which is not part of this paper). What are the key results that people are learning for the first time? The conclusion part mentions many interesting topics such as “variable interatomic distances and coordination numbers of amorphous alloys” but unfortunately they are not the focus of this study. The conclusion reads more like promoting the method (which is also well-known and used in the community) or the VAMPIRE software rather than showing important results.

  1. Related to the previous point but more on the study itself: the novelty of this study is not clear to me. I have difficulty extracting useful information from this study besides what is already known in the literature. While the composition dependent analysis of FexGd_{100-x} is done thoroughly, the results seem to be expected and the author has shown the past experiments to support this. My suggestion to the author is to think through the following questions to help you emphasize the merit of this work, rather than simply describing what has been done:

Q1. Does this work generate any new or surprising results that were not known in the literature?

Q2. Is my work promoting new findings or a new approach?

Q3. Since this is a simulation work, am I using the simulation to explain an experiment that people didn’t have a good understanding of its physics? Or am I exploring the parameter range beyond what was done in the experiment?

  1. The amorphous compound was simulated in BCC and HCP crystal structures with randomized atoms. Those two structure choices are not well justified for this study. From the context, it seems they were chosen because of the bulk crystal structures of Fe and Gd. However, the author has admitted the coordination number might play an important role in the magnetic properties under study. They didn't show evidence that those two structures are good substitutes for amorphous compounds synthesized in experiments and the reference [21] used the FCC structure in their study. This goes back to the issue of the core message of this study. If this were the first attempt to study amorphous Fe_xGd_{100-x} through atomistic simulations (which I don’t believe it is), then BCC and HCP structures are reasonable choices to start with. If the goal is to show how different coordination numbers can affect the magnetic properties, then the argument might be that using structures like BCC and HCP is the simplest way to vary the coordination number within the VAMPIRE software.

  1. One minor comment about figure 5. The leftmost image of the structure being simulated is difficult to see. The author should either use a zoomed in version where the exact structure is shown clearly or use a schematic figure (like the schematics on the left of Figure 1)

Round 2

Reviewer 2 Report

The revised manuscript is improved evidently and I also agree with the revision.

Author Response

The authors would like to thank the reviewer for the valuable comments and careful review.

Reviewer 3 Report

I appreciate the authors making an effort to revise the manuscript from the previous comments. In my opinion, the quality of the paper has greatly improved from those changes. Now I can easily see the messages of the manuscript and why a reader would be interested in reading it. The updated version now clearly states why such study was done (to show how atomistic simulations can be used to study the amorphous system and its advantages over first principle approaches), what conclusions are drawn from the study (the impact of exchange coupling strength and coordination number), whether the method should be trusted and its known limitations (comparison to experiments that was nicely summarized in Table 1). While some of the conclusions weren't a surprise to the community, I do see the merit of this manuscript as a justification and promotion of the approach used here, which can be of value to some researchers. I only have two very minor comments regarding some changes made in the revision:

  1. Some cells in Table 1 are highlighted with different colors (blue underline versus green underline). Why different colors? From the text, it seems the highlighted ones show better agreement with experimental works but doesn’t mention what different color means. If they were meant for different experimental work, it is not necessary because the reader can clearly see the different columns. Since they convey the same message, I would go with the same color.

  2. In the description of Table 1 (Page 9 line 281), the line says “... a slight decrease of Tc is observed in Gd reach region.” What is the “Gd reach region”? I thought about “Gd rich region” but the context seems to suggest it is describing the Fe rich region.
